# Humanization of the Reaction Specificity of Mouse Alox15b Inversely Modified the Susceptibility of Corresponding Knock-In Mice in Two Different Animal Inflammation Models

**DOI:** 10.3390/ijms241311034

**Published:** 2023-07-03

**Authors:** Marjann Schäfer, Florian Reisch, Dominika Labuz, Halina Machelska, Sabine Stehling, Gerhard P. Püschel, Michael Rothe, Dagmar Heydeck, Hartmut Kuhn

**Affiliations:** 1Department of Biochemistry, Charité—Universitätsmedizin Berlin, Corporate Member of Freie Universität Berlin and Humboldt-Universität zu Berlin, Charitéplatz 1, D-10117 Berlin, Germany; marjann.schaefer@googlemail.com (M.S.); reisch_florian@web.de (F.R.); sabine.stehling@charite.de (S.S.); dagmar.heydeck@charite.de (D.H.); 2Institute for Nutritional Sciences, University Potsdam, Arthur-Scheunert-Allee 114–116, D-14558 Nuthetal, Germany; gpuesche@uni-potsdam.de; 3Department of Experimental Anesthesiology, Charité—Universitätsmedizin Berlin, Corporate Member of Freie Universität Berlin and Humboldt-Universität zu Berlin, Hindenburgdamm 30, D-12203 Berlin, Germany; dominika.labuz@op.pl (D.L.); hmachelska@gmail.com (H.M.); 4Lipidomix GmbH, Robert-Roessle-Straße 10, D-13125 Berlin, Germany; michael.rothe@lipidomix.de

**Keywords:** eicosanoids, lipids, metabolism, fatty acids, inflammation, atherosclerosis

## Abstract

Mammalian arachidonic acid lipoxygenases (ALOXs) have been implicated in the pathogenesis of inflammatory diseases, and its pro- and anti-inflammatory effects have been reported for different ALOX-isoforms. Human ALOX15B oxygenates arachidonic acid to its 15-hydroperoxy derivative, whereas the corresponding 8-hydroperoxide is formed by mouse Alox15b (Alox8). This functional difference impacts the biosynthetic capacity of the two enzymes for creating pro- and anti-inflammatory eicosanoids. To explore the functional consequences of the humanization of the reaction specificity of mouse *Alox15b* in vivo, we tested *Alox15b* knock-in mice that express the arachidonic acid 15-lipoxygenating Tyr603Asp and His604Val double mutant of *Alox15b*, instead of the arachidonic acid 8-lipoxygenating wildtype enzyme, in two different animal inflammation models. In the dextran sodium sulfate-induced colitis model, female *Alox15b*-KI mice lost significantly more bodyweight during the acute phase of inflammation and recovered less rapidly during the resolution phase. Although we observed significant differences in the colonic levels of selected pro- and anti-inflammatory eicosanoids during the time-course of inflammation, there were no differences between the two genotypes at any time-point of the disease. In Freund’s complete adjuvant-induced paw edema model, *Alox15b*-KI mice were less susceptible than outbred wildtype controls, though we did not observe significant differences in pain perception (Hargreaves-test, von Frey-test) when the two genotypes were compared. our data indicate that humanization of the reaction specificity of mouse *Alox15b* (*Alox8*) sensitizes mice for dextran sodium sulfate-induced experimental colitis, but partly protects the animals in the complete Freund’s adjuvant-induced paw edema model.

## 1. Introduction

Inflammation is an adaptive response by the immune system of complex organisms, which is triggered by pathogens and/or tissue injury [1,2]. Although the complex mechanisms of inflammation are not completely understood, considerable progress has recently been made in improving our understanding of the molecular and cellular mechanisms of inflammatory diseases [2]. A well-controlled inflammatory reaction is beneficial for the organism as inflammatory stimuli are eliminated and tissue repair is initiated. However, when the immune system loses control [3] or inflammatory resolution is dysregulated [4], inflammation can become detrimental. Fortunately, in most cases, acute inflammation is terminated via inflammatory resolution [5], but this process may not be the end of inflammation [6]. After the resolution cascade is completed, there is further immunological activity that redefines what was previously termed restorative homeostasis [6]. For many years, it was believed that inflammatory resolution was passively initiated when the inflammatory stimuli disappeared [7]. However, more recent data indicate that inflammatory resolution is an active process characterized by targeted biosynthesis of pro-resolving mediators [8]. These mediators terminate recruitment of neutrophils into the inflamed tissue, induce neutrophil apoptosis, activate clearance of apoptotic cells by anti-inflammatory macrophages and initiate alternative macrophage polarization [9]. Consequently, the inflamed tissue is cleaned up, repair processes are initiated and tissue homeostasis is re-established.

Alternative macrophage polarization [10] is a key process in inflammatory resolution, and this process is associated with a switch in lipid mediator profiles [11,12]. When human peripheral monocytes are differentiated in vitro to pro-inflammatory M1 macrophages (culturing naive macrophages in vitro in the presence of colony stimulating factor and interferon gamma) and subsequently stimulated with toll-like receptor agonists, a mediator profile is formed, in which ALOX5 and COX2-derived eicosanoids are dominant [13]. In contrast, alternative macrophage polarization (culturing naive macrophages in the presence of colony stimulating factor and interleukin 4) strongly upregulates the biosynthesis of ALOX15/ALOX15B-derived eicosanoids [13]. Several arachidonic acid (AA) lipoxygenases (ALOX-isoforms), such as ALOX15 [14], ALOX15B [15], ALOX12 [16] and ALOX5 [17], are implicated in normal and/or alternative macrophage polarization, but the detailed roles of these enzymes are still matters of discussion.

ALOX-isoforms form a family of polyenoic fatty acid oxidizing enzymes, which convert these substrates to hydroperoxy derivatives [18,19,20,21]. In cells, these intermediates are rapidly reduced to form the more stable hydroxy compounds or further metabolized to create more complex products, such as pro-inflammatory leukotrienes [22] and/or pro-resolving mediators, such as resolvins, maresins and protectins [23]. The mouse genome involves seven functional *ALOX* genes, and most of these genes are localized in a joint *Alox* gene cluster on chromosome 11 [24]. In the human genome, an orthologous gene exists for each mouse *Alox* isoform, though the gene encoding for mouse Aloxe12 is a corrupted pseudogene in humans [25]. Although mouse and human ALOX orthologs share a high degree (>80%) of amino acid conservation, there are remarkable functional differences between several enzyme orthologs. For instance, mouse Alox15 catalyzes AA 12*S*-lipoxygenation [26], while the human ortholog produces 15*S*-HETE as a dominant AA oxygenation product [27]. Human ALOX15B converts AA almost exclusively to 15*S*-HETE [15], but the corresponding mouse ortholog (Alox15b), which is sometimes called 8-LOX, according to the AA based enzyme nomenclature, catalyzes AA 8*S*-lipoxygenation. The structural basis for the different reaction specificities of the two mammalian ALOX15B orthologs was explored in previous studies [25,28,29,30]. When Tyr603 and His604 of recombinant mouse *Alox15b* were mutated to the amino acids, which are present at these positions in the human ortholog (Asp and Val, respectively), the corresponding double mutant (Tyr603Asp and His604Val) catalyzed almost exclusive AA 15*S*-lipoxygenation [30]. When an inverse mutagenesis strategy was applied for human ALOX15B, the reaction specificity of this enzyme was strongly shifted in favor of AA 8*S*-lipoxygenation [30]. These data indicated that Tyr603 and His604 function as determinants of the reaction specificity of mouse and human ALOX15B.

We recently [31] created *Alox15b*-knock-in mice, which express the AA 15-lipoxygenating Tyr603Asp and His604Val double mutant of mouse *Alox15b* instead of the AA 8-lipoxygenating wildtype enzyme. These mice *(Alox15b*-KI or *Alox8*-KI) are viable, breed normally and female individuals do not show significant phenotypic alterations up to an age of 65 weeks. Here, we tested young *Alox15b*-KI mice in two different whole-animal inflammation models and found that *Alox15b*-KI mice lost significantly more bodyweight during the acute phase of dextran sodium sulfate (DSS)-induced colitis than outbred wildtype controls. In contrast, in the adjuvant-induced paw edema model, *Alox15b*-KI mice were partly protected from inflammation, as indicated by the lower degree of paw swelling. These data suggest a different patho-physiological role of *Alox15b* in two different whole-animal inflammation models.

## 2. Results

### 2.1. Alox15b Knock-in Mice Express an Arachidonic Acid 15-Lipoxygenating Alox15b Mutant Instead of the Arachidonic Acid 8-Lipoxygenating Wildtype Enzyme

Mouse *Alox15b* is an AA 8-lipoxygenating enzyme [32,33], which is expressed at high levels in phorbol ester-treated skin. In vitro mutagenesis (Tyr603Asp and His604Val exchange) altered the reaction specificity of the recombinant enzyme in favor of 15-HETE formation. To explore whether this mutagenesis strategy also worked in vivo, we created *Alox15b* knock-in mice (*Alox15b*-KI), which expressed an AA 15-lipoxygenating enzyme variant (Tyr603Asp and His604Val double mutant) instead of the AA 8-lipoxygenating wildtype enzyme. To test whether our in vivo mutagenesis strategy actually worked, we performed ex vivo activity assays using a homogenate of phorbol myristic acid (PMA)-treated tail epidermis as the enzyme source. We incubated aliquots of the homogenate supernatant with exogenous AA, prepared the formed conjugated dieses via RP-HPLC and further analyzed these products via combined normal phase/chiral phase HPLC. From Figure 1A, it can be seen that 12*S*-HETE was the major AA oxygenation product, yet the metabolic origin of this compound was not explored. In addition, smaller amounts of 8*S*-HETE were also detected, yet 15*S*-HETE was hardly formed. When an epidermis homogenate supernatant of *Alox15b*-KI mice was used as the enzyme source, 12*S*-HETE was also the major oxygenation product. However, significant amounts of 15*S*-HETE were also detected (Figure 1B), while 8*S*-HETE was absent. These ex vivo activity data are consistent with the functional consequences expected from our in vivo mutagenesis strategy.

### 2.2. In DSS Colitis Alox15b-KI Mice Experience a More Severe Loss of Body Weight and Recovered Less Rapidly

The DSS–colitis model is a frequently employed model of intestinal inflammation [34]. Since humanization of the reaction specificity of *Alox15b* might impact the biosynthetic capacity of this enzyme for pro- and/or anti-inflammatory eicosanoids, we tested the susceptibility of the *Alox15b*-KI mice and outbred wildtype controls in the DSS–colitis model.

For this purpose, female mice received 2% DSS in the drinking water for 5 consecutive days. On the sixth day of the experimental protocol, the DSS solution was replaced with normal drinking water, and the animals were allowed to recover for an additional 10 days. As major clinical readout parameters that characterize the severity of intestinal inflammation, we quantified the body weight kinetics and colon lengths. When wildtype animals received DSS in the drinking water, the mice gained some weight during the first 4 days of the treatment period, which is consistent with previous literature reports [35,36]. At day 5 of the experimental protocol, the animals started losing bodyweight, and at day 7, the lowest bodyweight values were reached. Afterwards, the animals recovered from intestinal inflammation and quickly reached their initial bodyweights at day 10 of the experimental protocol (Figure 2A).

For *Alox15b*-KI mice, the loss of bodyweight started earlier (after day 2 of the experimental protocol) and, as for the wildtype mice, the lowest bodyweight values were reached at day 7. Subsequently, the *Alox15b*-KI mice also recovered but with a lower rate. Most interestingly, between day 2 and day 15, the curve characterizing the body weight kinetics of the wildtype animals was consistently above the curve observed for the *Alox15b*-KI mice. These data suggest that the genetically modified mice were more sensitive to DSS treatment, and statistic evaluation of the body weight kinetics (two-way ANOVA, *p* = 0.001) indicated a significant difference between the two genotypes.

DSS-induced colitis involves restructuring the colon wall during the acute phase of inflammation, and these processes induce colon shrinkage. The degree of colon shrinkage has been suggested to mirror colitis severity [37]. We determined the degree of colon shrinkage to be the second clinical readout parameter, and we found that DSS treatment of wildtype mice induced significant colon shrinkage and the colon length normalized during the recovery period (Figure 2B). A similar colon shrinkage was observed for the *Alox15b*-KI mice, and these animals also recovered by day 8 of the experimental protocol. However, there was no significant difference between the two genotypes at either time point during the experimental protocol (Figure 2B).

Evaluation of histological cross-sections of colon tissue [38] of *Alox15b*-KI mice and wildtype controls (Figure 2C) revealed regular tissue structures for both genotypes at day 0 of the experimental protocol (no DSS treatment). In contrast, at the end of the acute inflammatory phase (day 5 of the experimental protocol), we detected clear indications of intestinal inflammation, such as neutrophil infiltration and mucosal erosion, in both genotypes. However, these changes were rather subtle, and we did not detect dramatic ulcerations of the mucosa. At day 15 of the experimental protocol, we still observed infiltration of neutrophils, but, otherwise, the structure of the mucosa looked rather normal. Once again, we did not observe dramatic differences between the two genotypes.

### 2.3. Oxilipin Profiles in Colon Tissues of Alox15b-KI Mice and Wildtype Controls during the Time-Course of DSS-Induced Colitis

Since the higher susceptibility of *Alox15b*-KI mice in the DSS–colitis model might be related to a different profile of pro- and/or anti-inflammatory eicosanoids, we next employed a lipidomic approach and quantified the steady state concentrations of 44 different eicosanoids in the colon tissue. For this purpose, we extracted the colon lipids, hydrolyzed the ester lipids and quantified the resulting oxygenated fatty acid derivatives via LC-MS. An overview of all quantified oxylipins and the detection limits for the different metabolites are given in Appendix A. When we summed up all oxylipins present in the colon tissues at the different time points during the experimental protocol (Figure 3), we observed that DSS treatment induced a significant increase in the levels of oxylipins from about 50 µg/g (no DSS treatment) to about 90 µg/g wet weight (7 days of DSS treatment). During the recovery period (8th day after DSS removal), the colonic oxylipin levels returned to the values observed in animals that did not receive DSS. However, there were no significant differences between the oxylipin levels detected for the two genotypes at any of the time points during the experimental protocol.

From Figure 3, it can be seen that the OH-PUFA content of normal and inflamed colon tissue varied between 50 and 80 µg/g tissue protein, and there were no differences between the two genotypes at any time point during the inflammation kinetics. Since we did not quantify the PUFA tissue content via LC-MS/MS, we could not calculate the hydroxy–PUFA/PUFA ratio as a measure of the degree of oxidative modification of the tissue lipids. However, in separate experiments, we extracted the total lipids from normal mouse tissue (liver, colon, kidney, brain), hydrolyzed the lipid extracts and analyzed the hydroxy–PUFA/PUFA ratio via RP-HPLC. Here, we found that this ratio varied between 0.01 and 0.1%, indicating that only minor shares of the PUFAs are present in normal tissue as hydroxylated derivatives. Corresponding data for inflamed colon tissue were not obtained.

When we quantified the relative shares of the different oxylipins, which contributed to the sum of the oxidized fatty acid derivatives, we found that the AA-derived hydroxy fatty acids were dominant (Figure 4). In fact, with more than 20 µg/g wet weight, 12-HETE was the major oxylipin detected in the colon tissue (Figure 4C), followed by 15-HETE (about 10 µg/g wet weight, Figure 4A) and 11-HETE (10 µg/g wet weight, Figure 4B). Other HETE isomers, such as 5-HETE (Figure 4F), 8-HETE (Figure 4D) and 9-HETE (Figure 4E), were detected at lower concentrations. For the 12-HETE levels (Figure 4C), a clear time course was observed during the experimental protocol. DSS treatment induced a significant increase in the colonic 12-HETE concentrations, which then returned to normal during the recovery period.

Similar significant differences were observed for *Alox15b*-KI mice (*p* = 0.0054 for no DSS vs. 7 days DSS and *p* = 0.0176 for 7 days DSS vs. 8 days after DSS removal). For the other HETE isomers, similar kinetics were observed, but here the differences did not always reach the levels of statistical significance (Appendix A). Interestingly, the 8-HETE levels of *Alox15b*-KI mice not treated with DSS were significantly (*p* = 0.0098) lower than the values obtained for outbred wildtype controls (Figure 4D). These data suggest that mouse *Alox15b* with its AA 8-lipoxygenating activity may have contributed to 8-HETE formation in colon tissue. On the other hand, we did not observe the concomitant increase in the colon 15-HETE levels (Figure 4A) that was expected from our in vivo mutagenesis strategy. In fact, we even observed significantly lower 15-HETE levels in the colon wall of *Alox15b*-KI mice (Figure 4A, left columns). This outcome may be related to the fact that the colonic 8-HETE levels (Figure 4D) were 20-fold lower than the 15-HETE levels (Figure 4A). In other words, the small amounts of 15-HETE formed by the humanized *Alox15b* may not be sufficient to elevate the colonic 15-HETE levels. Other biosynthetic pathways for 15-HETE, including AA auto-oxidation, may be more relevant for 15-HETE formation in the colon tissue than the humanized *Alox15b* pathway.

Next, we quantified the oxygenation products of 5,8,11,14,17-eicosapentaenoic acid (EPA) and found that 18-HEPA, which was previously detected in large amounts in human blood plasma [39], was only present in small amounts (Figure 5A). In vitro, this metabolite is only formed in trace amounts from EPA by recombinant human ALOX15 [40]; however, owing to the different reaction specificity of mouse *Alox15*, it is unlikely that this enzyme plays a major role in 18-HEPE formation.

As for the AA oxygenation products, the 12-hydroxy derivative (12-HEPE, Figure 5C) was present at the highest concentrations. However, compared to 12-HETE (20 µg/g wet weight), the 12-HEPE levels (0.7 µg/g wet weight) were more than one order of magnitude lower. 11-HEPE, 5-HEPE, 8-HEPE and 9-HEPE were present, albeit even at lower concentrations. For 15-HEPE and 12-HEPE, we observed similar kinetics to those of the AA oxygenation products during the time-course of inflammation. After the acute phase of inflammation, the colonic HEPE levels were increased, but they went back to normal during the resolution phase. For 18-HEPE, 11-HEPE, 8-HEPE, 9-HEPE and 5-HEPE, such kinetics were not observed. More detailed information (direction of alteration, *p*-values) on the oxygenated colonic EPA derivatives is provided in Appendix A. Most interestingly, in this study, we did not observe significant differences for any of the EPA metabolites when *Alox15b*-KI mice were compared to outbred wildtype controls.

4,7,11,13,16,19-Docosahexaenoic acid (DHA) is of nutritional relevance since, in mammals, it is synthesized only in small amounts [41,42]. Due to its high degree of unsaturation, this polyenoic fatty acid is very sensitive to auto-oxidation, but it also constitutes a suitable substrate for different ALOX-isoforms [40]. As major DHA oxygenation products, we identified 17-HDHA (Figure 6C) and 14-HDHA (Figure 6E) in colonic tissue, followed by 20-HDHA (Figure 6A), 13-HDHA (Figure 6D) and 4-HDHA (Figure 6J). Without DSS treatment, only small amounts of 17- and 14-HDHA were detected in the colon of both genotypes. After 5 days of DSS treatment, significantly elevated tissue concentrations of these metabolites were found, which were reduced to normal values after the recovery period (10 days after DSS removal). For other metabolites, similar kinetics were observed, though the differences did not always reach the level of statistical significance (Appendix A).

Although we detected subtle but statistically significant differences between the two genotypes for 20-HDHA, 16-HDHA, 11-HDHA, 7-HDHA, 8-HDHA and 4-HDHA in untreated colon, there was no convincing evidence in support of the assumption that humanization of the reaction specificity of *Alox15b* altered the pattern of DHA oxygenation products. More detailed information (direction of alteration, *p*-values) on the oxygenated colonic DHA derivatives is provided in Appendix A.

Linoleic acid (LA) is one of the most abundant polyenoic fatty acids in mammalian tissues, and we identified large amounts of 13-HODE (Figure 7A) and 9-HODE (Figure 7B) in the lipid extracts of normal and inflamed colon. Since similar amounts of 13- and 9-HODE were detected, it appears unlikely that any of the mouse *Alox*-isoforms significantly contributed to the biosynthesis of these metabolites. There were no systematic alterations in the tissue concentrations of these two metabolites during the time-course of the inflammation reaction, and we did not observe significant differences between *Alox15b*-KI mice and outbred wildtype controls.

Alpha-Linolenic acid (ALA) occurs in much lower amounts in mammalian tissues, and we detected smaller tissue concentrations of 13-HOTrE (Figure 7C) and 9-HOTrE (Figure 7D). As for the LA derivatives, we observed neither systematic changes in the tissue concentrations of these metabolites during the time course of inflammation nor significant differences between the two genotypes. Similar conclusions were drawn for the oxygenation metabolites of 8,11,14-eicosatrienoic acid (Figure 7E–G). Here, we found that in the absence of DSS treatment, significantly lower amounts of 8-HETrE were present in the colon of *Alox15b*-KI mice compared to outbred wildtype controls (Figure 7F). Similar results were obtained for 8-HETE when the AA oxygenation products were quantified (Figure 4D). This finding is an interesting observation since these data are consistent with the functional consequences that we predicted via our genetic manipulation. However, as for the AA metabolites, we did not observe an anti-parallel drop in the 15-HETrE concentrations (Figure 7G).

Finally, we quantified the tissue steady state concentrations of more complex oxylipins, including leukotriene B4 (LTB4), as well as different resolvin, maresin and neuroprotection isomers. LTB4 is one of the most powerful pro-inflammatory eicosanoids [43], and we observed (Figure 8A) only small amounts of this metabolite in colon tissue of untreated wildtype mice. During the acute phase of inflammation, LTB4 tissue concentrations were strongly increased (*p* < 0.0001) but returned to normal during the recovery period (*p* = 0.0004 for 5 days of DSS treatment vs. 10 days after DSS removal). Almost identical kinetics were observed when *Alox15b*-KI mice were analyzed. Once again, highly significant differences (Appendix A) were detected in the colon levels of LTB4 at the different time points (*p* < 0.0001 for no DSS vs. 5 days of DSS treatment; *p* < 0.0001 for 5 days DSS treatment vs. 10 days after DSS removal).

For the LTB4 metabolite 18-carboxy-dinor LTB4 (Figure 8B) and LTB3 (Figure 8C), similar metabolite kinetics were observed, but here, the differences between the time points of inflammation did not always reach the level of statistical significance (Appendix A). Neuroprotectin 1 (NPD1, 10*R*,17*S*-dihydroxy-4*Z*,7*Z*,11*E*,13*E*,15*Z*,19*Z*-docosahexaenoic acid), which is a dihydroxy derivative of DHA, exhibits neuroprotective properties [44,45] and has been classified as a special pro-resolving mediator (SPM) [46]. We detected small amounts of NPD-1 in the colon of untreated wildtype mice (Figure 8D), and the tissue concentration was strongly increased after 5 days of DSS treatment (*p* < 0.0001). After the recovery period, the NPD-1 levels went back to normal (*p* = 0.001). For *Alox15b*-KI mice, similar alterations were observed (*p* < 0.0001 for no DSS vs. 5 days of DSS treatment; *p* = 0.0220 for 5 days of DSS treatment vs. 10 days after DSS removal). Thus, during the time course of inflammation, the NPD-1 kinetics were very similar to the kinetics observed for the pro-inflammatory LTB4 (Figure 8A). Similar kinetics were observed for the dihydroxy SPMs maresin-2 (13*R*,14*S*-dihydroxy-4*Z*,7*Z*,9*E*,11*E*,16*Z*,19*Z*-docosahexaenoic acid, Figure 8E) and resolving D5 (7*S*,17*S*-dihydroxy-4*Z*,8*E*,10*Z*,13*Z*,15*E*,19*Z*-docosahexaenoic acid, Figure 8F), and the *p*-values are summarized in Appendix A. For neither of the complex SPMs quantified in this study did we observe significant differences between the two genotypes.

PGB2 and PGB3 are non-enzymatic dehydration products of PGE2 and PGE3, and their tissue concentrations may mirror the levels of the biologically active but less stable PGE2 and PGE3. When we quantified these metabolites in the colon tissues during the time course of experimental colitis, we detected large amounts of PGB2 but much lower amounts of PGB3. However, there were no time-dependent alterations in the tissue concentrations of these metabolites during the time-course of inflammation, and we did not observe major differences between the two genotypes. 15-deoxy-PGJ2-12,14 is the dehydration product of PGD2 that exhibits its biological functions via the PGD2 receptors (DP1 and DP2). It has been identified as an endogenous ligand required for the intranuclear receptor PPARgamma [47]. This property is responsible for its anti-inflammatory functions, and the mechanistic details of the receptor–ligand interaction were explored in prior studies [48,49]. When we quantified the tissue concentration of this metabolite during the time course of inflammation, we observed similar profiles to those of LTB4, NPD-1, Maresin-2 and RvD5. Without DSS, we only detected small amounts of this metabolite in the colon tissue. During the acute phase of inflammation, the tissue levels were increased but returned to normal after the recovery phase (Figure 8I), and the *p*-values indicating the statistical significance of the time-dependent differences are given in Appendix A.

We also quantified the colon tissue concentrations of other dihydroxy PUFAs, such as 5*S*,12*S*-diHETE, 8*S*,15*S*-diHETE and 10*S*,17*S*-diHDHA, in the colon tissue of wildtype and *Alox15b*-KI mice. Here, we found (Table 1) that in wildtype mice, these dihydroxylated metabolites followed similar kinetics to those of leukotriene B4, neuroprotection-1, maresin-2 and resolvin-D5 (Figure 8) during the time-course of inflammation. Their tissue concentrations were low before DSS administration, but they strongly increased five days after the start of DSS application. Interestingly, after the recovery period, the metabolite concentrations did normalize once again. The differences between the day 0 and day 5 groups, on one hand, and the day 5 and day 15 groups, on the other hand, were statistically significant (*p* < 0.05) for most metabolites. In contrast, no significant differences were observed when the day 0 and day 15 groups were compared. The metabolite kinetics suggested that the different dihydroxy PUFAs were formed during the acute phase of inflammation, as well as that they were largely removed during the resolution phase. Moreover, we found that the genotype of the mice hardly impacted the metabolite kinetics. In *Alox15b*-KI mice, we also observed increased dihydroxy PUFA concentrations during the acute inflammation period, which was followed by a subsequent decline in the frame of inflammatory resolution. In most cases, the metabolite concentrations in the colon tissue of *Alox15b*-KI mice were similar to those detected in the colon of outbred wildtype control animals (Table 1).

We also attempted to quantify a number of additional dihydroxy derivatives of EPA (RvE2 = 15*S*,18*R*-diHEPE, RvE4 = 5*S*,15*S*-diHEPE, LTB5 = 5*S*,12*R*-diHEPE), but these metabolites were below the detection limits used by our analytical method. Similarly, we attempted to quantify maresin-1 (Mar1), but this metabolite could not be detected in the colon tissue. 9*S*,13*S*-diHOTrE was not a constituent part of our analytical scheme, and thus, we cannot comment on the colon concentrations of this metabolite.

In addition to these dihydroxy SPMs, we also attempted to quantify the following trihydroxy-SPMs: RvD1, RvD2, RvD3, RvD4, LxA4 and LxB4. Unfortunately, none of these metabolites could be quantified in the normal or inflamed colon of either genotype (Appendix A). If present, the tissue concentrations were below the detection limits of our analytical procedure (Appendix A).

### 2.4. Alox15-Knock-in Mice Show Reduced Paw Swelling in the CFA-Induced Paw Inflammation Model

To test the *Alox15b*-KI mice in a second animal inflammation model, we used the paw edema system [50]. For this purpose, we injected complete Freund’s adjuvant into a hind paw of the mice, and after two days, we determined three clinical readout parameters (paw volume, paw removal latency after thermic stimulation (Hargreaves test) and paw withdrawal threshold after mechanic stimulation (von Frey test)). As a control, the same volume of isotonic saline was injected into the contralateral hind paw.

From Figure 9A, it can be seen that injection of PBS into the hind paw did not alter the paw volume compared to the untreated paws (blank, BL) of wildtype mice. However, injection of complete Freund’s adjuvant led to a significant increase in the paw volume, indicating the formation of an inflammatory paw edema. Similar alterations were observed for *Alox15b*-KI mice (Figure 9B). When we compared the degree of paw swelling between *Alox15b*-KI mice and outbred wildtype controls, we observed more pronounced paw swelling for the wildtype animals (Figure 9C). These data suggest that *Alox15b*-KI mice were partially protected from adjuvant-induced paw swelling under our experimental conditions.

Inflammation not only induces tissue swelling (edema), but also induces pain. Inflammation-induced pain can be quantified via a number of complex assay systems [51], and we employed the Hargreaves [52] and von Frey tests [53]. When we compared the infrared light-induced paw withdrawal latency (Hargreaves test) of untreated wildtype mice with that of NaCl injected controls, we did not see any difference (Figure 10A). However, as expected, injection of complete Freund‘s adjuvant significantly reduced the paw withdrawal latency (Figure 10A). Similar effects were observed for *Alox15b*-KI mice (Figure 10B). Finally, we compared the paw withdrawal latencies of adjuvant-treated *Alox15b*-KI mice with those of the adjuvant-treated wildtype controls (Figure 10C), but did not observe significant differences. These data suggest that humanization of the reaction specificity of mouse *Alox15b* does not impact the sensitivity of mice to heat-induced pain.

Finally, we employed the von Frey assay to test whether the humanization of the reaction specificity of *Alox15b* might impact the paw withdrawal threshold when paws were stimulated mechanically. Once again, we first compared the threshold value of untreated wildtype mice with that of saline injected individuals, but did not observe any differences (Figure 10D). As in the Hargreaves test, injection of complete Freund’s adjuvant significantly increased the pain sensitivity of wildtype animals (Figure 10D). Similar effects were observed in *Alox15b*-KI mice (Figure 10E). Finally, we compared the paw withdrawal threshold of adjuvant-treated *Alox15b*-KI mice with that of outbred wildtype controls (Figure 10F). Here, we observed a lower threshold for *Alox15*-KI mice, but this difference did not reach the level of statistical significance. Taken together, from the data obtained via the Hargreaves and von Frey assays, we can conclude that humanization of the reaction specificity of mouse Alox15b hardly impacts the sensitivity of mice to inflammation-induced pain.

## 3. Discussion

### 3.1. Alox15-KI Mice Lose Significantly More Body Weight during the Time Course of DSS-Induced Colitis

Human ALOX15B oxygenates AA almost exclusively to 15*S*-HETE [15], but the corresponding mouse enzyme (Alox15b) forms 8*S*-HETE as a major mono-hydroxylated AA derivative [32,33]. Since AA 15-lipoxygenating enzymes may more effectively convert ALOX5-derived precursors into pro-resolving lipoxins than AA 8-lipoxygenating enzymes, it was expected that *Alox15b*-KI mice, which express the AA 15-lipoxygenating Alox15b Tyr603Asp and His604Val double mutant, might develop less intense inflammatory symptoms in animal inflammation models. To test this hypothesis, we first induced experimental colitis [34,37] in *Alox15b*-KI mice and outbred wildtype controls and quantified the intensity of inflammation using two independent clinical readout parameters. By quantifying the body weight kinetics, we found that *Alox15b*-KI lost significantly more body weight (Figure 2A). On the other hand, we did not detect significant differences between the two genotypes when the degree of colon shrinkage was quantified as a readout parameter of the inflammatory response (Figure 2B). Moreover, despite evaluating the histological cross-sections of *Alox15b*-KI mice and wildtype controls, we did not observe significant differences between the two genotypes when we quantified the inflammatory alterations [38]. In fact, when we quantified the degree of infiltration by inflammatory cells, the degree of epithelia changes and the overall structure of the mucosa, we did not observe significant differences between the two genotypes at any time point during the experimental protocol (Figure 2C). These histological data suggested that the degree of intestinal inflammation was not significantly different between the two genotypes [34]. This conclusion was consistent with our lipidomic data, indicating that colonic levels of fatty acid oxygenation products (Figure 3) and pro-inflammatory mediator leukotriene B4 (Figure 8A) were not significantly different for the two genotypes at day 7 of the experimental protocol. Taken together, these results suggest that DSS treatment more severely compromises the intestinal water barrier of *Alox15b*-KI mice than that of wildtype controls, and this conclusion is consistent with the observed bodyweight kinetics. However, the degree of inflammation was not significantly impacted via humanization of the reaction specificity of Alox15b.

### 3.2. Humanization of Alox15b Reaction Specificity Does Not Induce Pronounced Alterations in the Oxylipin Profile in Normal and Inflamed Colon Tissues

To explore whether humanization of the reaction specificity of mouse *Alox15b* impacts the oxylipin profile in colon tissue, we performed lipidomic analyses and quantified the tissue concentrations of more than 40 different oxygenated PUFA derivatives [54] at different time points of the inflammation reaction (Figure 4, Figure 5, Figure 6, Figure 7 and Figure 8). Here, we found that the tissue concentrations of 8-HETE (Figure 4D) and 8-HTrETE (Figure 7F) in the colon of untreated mice were significantly reduced in *Alox15*-KI mice, and these data are consistent with the functional consequences expected based on our genetic manipulation [31]. For 8-HEPE (Figure 5F), we observed a similar trend, but here the difference did not reach a level of statistical significance. These data suggested that while *Alox15b* may contribute a small share to the formation of endogenous 8-hydroxy fatty acids in untreated colon tissue, the majority of these metabolites originate from alternative oxidation reactions. A second functional consequence that we expected from our genetic manipulation [31] was an anti-parallel increase in the endogenous 15-HETE, 15-HEPE and 15-HETrE levels, which we did not observe in our analysis. Our failure to find elevated 15-hydroxy PUFA levels in the colon of *Alox15b*-KI mice compared to wildtype controls may have several reasons, and two of those reasons may be discussed: (i) the 15-hydroxy PUFA levels are much higher than those of the 8-hydroxy derivatives, and the relatively small increase that was expected as consequence of our genetic manipulation may not be visible on top of the high background levels; and (ii) the 15-hydroperoxy derivatives formed by the humanized *Alox15b* may quickly be further metabolized to secondary products, which were not profiled in our analyses.

Another interesting aspect of our lipidome analyses was the observation that the tissue concentrations of the pro-resolving mediators Mrs-2 and RvD5 were at least one order of magnitude lower than that of the pro-inflammatory LTB4 (Figure 8). Of course, the tissue concentration of a mediator does not adequately mirror its bioactivity. However, considering the previous observation that LTB4 is one of the most powerful pro-inflammatory mediators [55,56,57], the specific activity of Mrs-2 and RvD5 [58,59] must be very high to compensate the pro-inflammatory effects of LTB4. Interestingly, NPD-1 was found in similar concentrations to those of LTB4 at day 7 of our experimental protocol, and, thus, the pro-resolving effect of this SPM might counteract the pro-inflammatory effect of LTB4 if the specific activities of the two metabolites are comparable.

The colon levels of 5-HETE and 9-HETE were significantly reduced in *Alox15b*-KI mice compared to wildtype controls (Figure 9E,F). A similar significant reduction was observed for the principle *alox5* products of DHA, 4-HDHA and 7-HDHA (Figure 6H,J). These results present the interesting question of whether *Alox15b* might regulate either the expression or the catalytic activity of *alox5*. To the best of our knowledge, such a regulatory response has not been described, but for cells expressing the two enzymes (inflammatory cells), such functional interplay might be possible. On the other hand, no studies previously explored before which share of the colonic 5-HETE levels may be related to the catalytic activity of *Alox5*. DSS–colitis experiments with *Alox5^−/−^* mice [60] in connection with lipidomic analyses would help to answer this question. For 9-HETE and the corresponding DHA metabolites 8-HDHA and 11-HDHA, for which we also analyzed reduced levels in untreated colons of *Alox15b*-KI mice (Figure 6G,I), the situation was somewhat different. These metabolites were not formed as major AA or DHA oxygenation products by any of the seven mouse Alox-isoforms and, thus, in mice, while in humans, these metabolites might be biosynthesized via alternative pathways, most probably auto-oxidation. Since the metabolic origin of these compounds has not been identified, it remains unclear through which mechanism humanization of the reaction specificity of mouse *Alox15b* might impact 9-HETE formation.

In our lipidomic studies, we analyzed the kinetic profiles of some putative pro-resolving metabolites (Figure 8), such as neuroprotectins (NPD-1, NPDx), maresin-2 (Mrs-2) and resolvin-D5 [5,46], during the time course of DSS-induced colitis. Here, we found low colon levels in untreated mice of either genotype. Five days after DSS-treatment, we observed a strong increase in metabolite concentrations, which returned to normal during the recovery period. A similar time course was observed for the classical pro-inflammatory mediator leukotriene B4. Due to the similar kinetics of these four complex oxylipins, we might conclude that NPD-1, Mrs-2 and RvD5 might also function as pro-inflammatory mediators in this particular model on inflammation. There is, however, an alternative explanation. When colonic inflammation is induced via DSS administration, the inflammatory response might instantaneously be followed by a resolving response, and, thus, pro-inflammatory and pro-resolving mediators might be synthesized more or less simultaneously. More comprehensive kinetic experiments (analyzing the colon concentrations of these metabolites in shorter time intervals during the acute phase of inflammation) are required to explore this interesting phenomenon in more detail.

### 3.3. Humanization of Alox15b Reaction Specificity of Alox15b Partly Protected Mice in the Paw Edema Inflammation Model but Does Not Impact Pain Perception

When we used the Freund’s adjuvant-induced paw edema model and quantified the degree of paw swelling as a clinical readout parameter for the severity of the inflammatory reaction, we found that *Alox15b*-KI mice developed a less intense edema than outbred wildtype control animals (Figure 9C). In other words, humanization of the reaction specificity of *Alox15b* partly protected these mice from inflammation. The molecular mechanisms for this protection have not been explored, but they may be related to the above-described down-regulatory effect of the *Alox5* pathway. As indicated in Figure 4F and Figure 6H,J, lower concentrations of *Alox5* products (5-HETE, 4-HDHA, 7-HDHA) were present in the colon tissue of untreated *Alox15b*-KI mice than in wildtype controls. Although these data do not prove a regulatory interplay between *Alox5* and *Alox15b*, they at least suggest that humanization of the reaction specificity of *Alox15b* might downregulate the catalytic activity of *Alox5*. Since leukotrienes play an important role in the pathogenesis of paw edema, partial suppression of edema formation would actually be expected. Whether this mechanistic scenario is, indeed, the reason for the observed protective effect must be explored in future experiments, in which the *Alox15b*-KI mice can be employed as suitable in vivo models.

Inflammation is always paralleled by pain [1], which can be quantified using different experimental setups. We employed two different assay systems (Hargreaves test, von Frey test) to quantify the intensity of inflammation-induced pain, but we did not observe significant differences between *Alox15b*-KI mice and outbred wildtype controls (Figure 10C,F). Although the paw withdrawal threshold (von Frey test, Figure 10F) for *Alox15b*-KI mice was lower than the corresponding value for wildtype controls, the difference did not reach the level of statistical significance (*p* = 0.069). Thus, humanization of the reaction specificity of *Alox15b* hardly impacted pain perception.

## 4. Materials and Methods

### 4.1. Chemicals

The chemicals used for the ex vivo activity assays were purchased from the following vendors: PBS (without Ca^2+^ and Mg^2+^) from PAN Biotech (Aidenbach, Germany); HPLC solvents from Fisher Scientific GmbH, Schwerte, Germany); HPLC standards of 15R/S-HETE, 12R/S-HETE, and 8-R/S-HETE from Cayman Chemicals (distributed by Biomol GmbH, Hamburg, Germany); and phorbol myristic acid (PMA) from Merck (Darmstadt, Germany). Dextran sulfate sodium (DSS, molecular weight = 36,000–50,000) was purchased from ICN Biomedicals (Irvine, CA, USA). The chemicals used for pain assessment (incomplete Freund’s adjuvant and desiccated *Mycobacterium butyricum*) were obtained from Fisher Scientific GmbH (Schwerte, Germany). The plethysmometer model 37140 was purchased from Ugo Basile (Gemonio, Italy). Von Frey cages (Model 410) and Hargreaves cages (Model 336) were purchased from IITC Life Sciences (Woodland Hills, LA, USA).

### 4.2. Animals

To create *Alox15b*-KI mice, we used the Crispr/Cas9 strategy and carried out in vivo mutagenesis, meaning that the resulting individuals expressed the AA 15-lipoxygenating *Alox15b* Tyr603Asp and His604Val double mutant instead of the AA 8-lipoxygenating wildtype enzyme. More detailed description of the mutagenesis strategy and a basic characterization of these animals were given in a previous paper [31]. For this study, homozygous *Alox15b*-KI mice were mated with wildtype C57Bl/6 mice, and the resulting heterozygous allele carries were intercrossed to obtain homozygous *Alox15b*-KI mice and homozygous wildtype controls. Founder animals were separately intercrossed, and colonies of *Alox15b*-KI mice and outbred wildtype controls were established. Each individual used for the described experiment was genotyped by sequencing the corresponding region of the *Alox15b* gene locus.

### 4.3. Ex Vivo Activity Assay of Mouse Alox15b

In mice, Alox15b is expressed at high levels in phorbol ester treated skin [32,33]; thus, we carried out ex vivo activity assays using PMA-treated mouse tail epidermis as an enzyme source. To obtain the skin, we sacrificed four *Alox15b*-KI mice and four outbred wildtype controls under anesthesia, amputated their tails and incubated them for 2 h in PBS containing 5 µM PMA. The epidermis was prepared, and the tissue was cut into small pieces with a pair of scissors and homogenized in 1 mL of PBS using a Fast-Prep-24 sample preparation system (MP Biomedicals, Irvine, CA, USA). Cell debris were spun down, and an aliquot of the homogenate supernatant was incubated in 1 mL PBS containing 100 µM AA for 30 min at room temperature. The reaction products were reduced via the addition of 1 mg solid sodium borohydride, the sample was acidified via the addition of 35 µL acetic acid and lipids were extracted twice with 1 mL of ethyl acetate. The extracts were combined, the solvent was evaporated under vacuum and the remaining lipids were reconstituted in 250 µL of acetonitrile. After vortexing, 250 µL of water and 5 µL of acetic acid were added, the sample was centrifuged to remove debris and 300 µL of the supernatant were injected through RP-HPLC analysis for preparation of the conjugated dienes formed during the incubation period. For this purpose, a Shimadzu instrument (LC20 AD) equipped with a diode array detector (SPD M20A) was used, and the hydroxy fatty acids were separated using a Nucleodur C18 Gravity column (Macherey-Nagel, Düren, Germany; 250 × 4 mm, 5 μm particle size) coupled with a guard column (8 × 4 mm, 5 μm particle size). A solvent system consisting of acetonitrile:water:acetic acid (70:30:0.1, by vol) was employed at a flow rate of 1 mL/min, and analytes were eluted isocratically at 25 °C. The conjugated dienes formed during the incubation period were prepared, the solvents were removed and the remaining lipids were reconstituted in 200 µL of hexane containing 0.1% acetic acid. To resolve the hydroxy fatty acid isomers formed during the incubation period, combined normal phase/chiral phase HPLC was carried out. For this purpose, a Chiralpak AD-H column (4.6 × 250 mm, 5 µm particle size, Daicel, Osaka, Japan) was connected to a Nucleosil pre-column (4.6 × 30 mm, 5 µm particle size, Macherey-Nagel, Düren, Germany), and the analytes were eluted isocratically using a solvent system consisting of n-hexane/methanol/ethanol/acetic acid (96/3/1/0.1, by vol) at a flow rate of 1 mL/min. The absorbance at 235 nm was monitored, and the retention times of authentic standards were indicated above the chromatographic traces.

### 4.4. Dextran Sulfate Sodium (DSS) Induced Experimental Colitis

DSS-induced experimental colitis is a frequently employed whole animal colitis model that closely mirrors the patho-physiological aspects of human intestinal inflammation [34,37]. In this model, intestinal inflammation is induced via oral application of DSS in the drinking water, and the loss of body weight can be determined as a major clinical readout parameter [34,37]. Since the extent of intestinal inflammation depends on the percentage of DSS present in the drinking water [34,37], we carried out two preliminary experiments (using 3 female wildtype mice for each experiment) to test the appropriate DSS concentrations (1.5% and 2%) in the drinking water. In these experiments, we found that the presence of 1.5% DSS only induced subtle inflammatory symptoms. In contrast, the presence of 2% DSS in the drinking water induced medium-degree inflammatory symptoms, and, thus, we decided to perform the main experiments using this DSS concentration.

For the main experiment, we randomly selected 12 female *Alox15b*-KI mice (12–15 weeks) and 12 outbred wildtype control mice of corresponding age. For each genotype, two cages were set up, and each cage involved the animals of one experimental group. Cage A included 6 wildtype mice that represented the acute inflammation phase, cage B included 6 wildtype mice that tested the inflammatory resolution phase, cage C included 6 *Alox15b*-KI mice that tested the acute inflammation phase, and cage D involved 6 *Alox15b*-KI mice that tested the inflammatory resolution phase. At day 0 of the experimental protocol, the body weights of all animals were determined, and the mean was set to 100%. After weighting, the mice received 2% DSS in the drinking water for 5 consecutive days, and the body weights were determined each day in the morning. After 5 days, the animals of cages A (wildtype acute inflammation phase) and C (*Alox15b*-KI acute inflammation phase) were sacrificed, and the colons were prepared and washed with PBS. For cages B and C, the DSS containing drinking water was replaced with normal drinking water, and the animals were allowed to recover from intestinal inflammation for 10 days. Unfortunately, three animals in cage D (*Alox15b*-KI, inflammatory resolution) had to be euthanized at day 5 of the experimental protocol because of intense inflammatory symptoms. Thus, to characterize inflammatory resolution, only 3 *Alox15b*-KI animals remained.

After preparation and washing, the colons were divided into four segments of similar length. The proximal segment was transferred to a 4% paraformaldehyde solution for later preparation of histological cross sections. The distal segment was incubated for 30 min in RNAlater solution and then shock-frozen in liquid nitrogen for future RNA preparation and gene expression studies. The two middle colon sections were shock-frozen in liquid nitrogen for later tissue lipid (oxylipidome profiles) and tissue protein extraction (immunoblotting of relevant gene products).

### 4.5. Solid Phase Tissue Lipid Extraction

After thawing, about 10 mg of colonic tissue was added to 490 µL of water. Next, 10 µL of an internal standard solution (LTB4-d4, 20-HETE-d6, 15-HETE-d8, 13-HODE-d4, 14,15-DHET-d11, 9,10-DiHOME-d4, 12,13-EpOME-d4, 8,9-EET-d11 or PGE2-d4; 10 ng/mL each) and 5 µL of a methanolic solution of tert-butyl hydroxy toluene (1 mg/mL) were added. Afterwards, 300 µL of water, 100 µL of 10 M NaOH solution and 500 µL methanol were added. The sample was incubated for 30 min at 60 °C (alkaline hydrolysis of the cellular ester lipids), put on ice for 5 min and acidified with 100 µL of 58% acetic acid. Afterwards, 2 mL of 0.1 M phosphate buffer was added, and the pH was adjusted to 6.0 through addition of small aliquots of acetic acid of NaOH. A 50 µL aliquot was removed for later determination of the protein concentration. The rest of the sample was centrifuged, and the protein-free supernatant was used for solid-phase lipid extraction on a combined reverse phase–ion exchange Agilent Bond-Elut-Certify II cartridge (200 mg, Agilent Technologies, Santa Clara, CA, USA). Before sample application, the cartridge was first rinsed with 3 mL of methanol and then conditioned with 3 mL or 0.15 M phosphate buffer, recording a pH of 6.0. After sample application, the cartridge was rinsed with 3 mL of a 1:1 mixture of water and methanol. Elution of the tissue lipids was carried out via rinsing the cartridge with a 74:25:1 (by vol) mixture of ethyl acetate:n-hexane:acetic acid. The solvents were evaporated under a stream of nitrogen, and the remaining lipids were reconstituted in 100 µL of a 3:2 mixture (by vol) of methanol and water.

### 4.6. LC-MS/MS Based Oxylipidome Analyses

To explore the patterns of the colon oxylipins, we quantified the amounts of more than 40 different oxygenated PUFA derivatives [54]. LC-MS/MS was performed using an Agilent 1290/II LC-MS system consisting of a binary pump, an autosampler and a column oven (Agilent Technologies, Waldbronn, Germany). For chromatographic separation, we employed an Agilent Zorbax Eclipse C_18_ UPLC column (150 × 2.1 mm, 1.8 µm particle size). The analytes were eluted at 30 °C with a solvent system that was mixed using two solvent stock solutions: Stock A: Water containing 0.05% acetic acid; and Stock B: 1:1 mixture (by vol.) of methanol:acetonitrile [31]. The HPLC system was connected with a triple quadrupole MS system (Agilent 6495 System, Agilent Technologies, Santa Clara, CA, USA) that was run in negative electrospray ionization mode. Each metabolite was detected simultaneously via two independent mass transitions. Experimental raw data were evaluated using the Agilent Mass-Hunter software package, version B10.0. For all metabolites, individual calibration curves were set up, and the lower detection limits were determined (Appendix A).

### 4.7. Complete Freund’s Adjuvant Induced Paw Inflammation Model

The experiment was approved by the local Animal Care Committee (Landesamt für Gesundheit und Soziales, Berlin, Germany) and registered under permission number G 0296/18. Mice were kept in groups of 3 individuals per cage, with free access to food and water, under environmentally controlled conditions (12 h light/dark cycle, light on at 7:00 h; 22–24 °C; humidity 60–65%). In total, 10 male wildtype and 10 male *Alox15*-KI mice were used in this experiment. Animals were adapted to handling and the test cages twice per day for 6 consecutive days. On day 7, the basal values for all tests were determined, (paw volume, von Frey- and Hargreaves test) and, afterwards, i. pl. injections were performed. For the Left hind paw, injection of 20 µL 0.9% NaCl occurred, while for the right hind paw, injection of 20 µL *M. butyricum* in Freund’s adjuvant containing 50 µg *M. butyricum* occurred. Animals were monitored on day 8 according to the score sheet established in agreement with the Animal Welfare Administration. On day 9 (2 days after FCA injection), animals were tested via the von Frey and Hargreaves tests, and the paw volumes were determined. Von Frey and Hargreaves tests were carried out as described in [61].

### 4.8. Statistics and Data Presentation

Experimental raw data were checked for normal distribution using the Anderson–Darling, D’Agostino–Pearson, Shapiro–Wilk and Kolmogorov–Smirnov tests. According to these tests, several data sets did not pass the normality test, and, in these cases, we applied non-parametric *t*-test variations for statistical evaluations. The Wilcoxon test was used as a paired *t*-test, and the Mann–Whitney U test was used as an unpaired *t*-test. Statistical analyses were carried out via the GraphPad Prism software package, version 8.4.3 for windows (GraphPad Software, San Diego, CA, USA). This software was also employed to generate the images.

## 5. Conclusions

Humanization of the reaction specificity of mouse *Alox15b* (*Alox8*) sensitizes mice for dextran sodium sulfate-induced experimental colitis, while it partly protected the animals in the complete Freund’s adjuvant-induced paw edema model. These data suggest that *Alox15b* exhibits variable functionality in the pathogenesis of inflammation depending on the animal inflammation model. The protective effect in the paw edema model may not be related to the formation of pro-resolving lipid mediators.

## Figures and Tables

**Figure 1 ijms-24-11034-f001:**
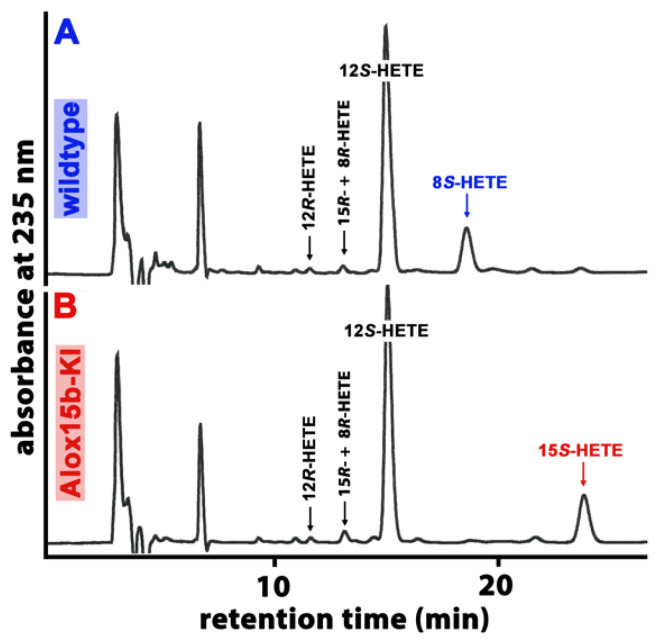
Ex vivo activity assays performed using homogenates of PMA-treated mouse skin as an enzyme source. The epidermis of phorbol myristate (PMA)-treated mouse tails were prepared from *Alox15b*-KI mice and outbred wildtype controls, they were homogenized and the homogenate supernatant was incubated in vitro with 100 µM AA (see Section 4). After lipid extraction, the conjugated dienes formed were prepared via RP-HPLC and further analyzed via combined NP/CP-HPLC, as described in Section 4. The absorbance at 235 nm was recorded, and the retention times of authentic standards are indicated above the chromatographic traces. (**A**) wildtype mice, (**B**) *Alox15b*-KI mice. Three different individuals of either genotype were analyzed, and representative NP/CP-HPLC chromatograms are shown.

**Figure 2 ijms-24-11034-f002:**
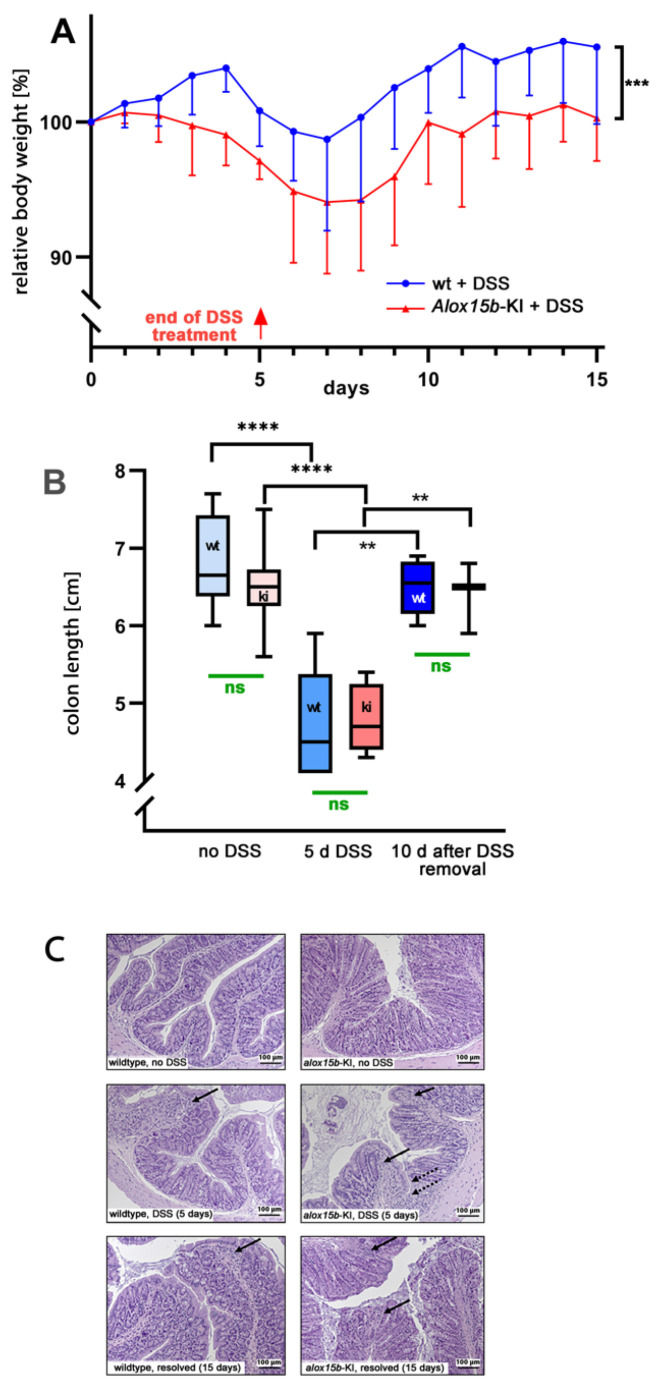
DSS-induced colitis in *Alox15b*-KI mice and outbred wildtype controls. Colonies of *Alox15b*-KI mice expressing an *Alox15b* mutant with humanized reaction specificity and outbred wildtype controls were established (see Section 4), and these animals were tested in the DSS-induced experimental colitis model. The experimental approach, animal grouping and quantification of the readout parameters are explained in detail in Section 4. (**A**) Body weight kinetics of *Alox15-*KI mice and outbred wildtype controls. Statistical analysis of the experimental raw data was performed via a two-way ANOVA; *** *p* < 0.001. (**B**) Colon lengths determined at different time points of the experimental protocol. Statistical evaluation of the experimental raw data was carried out via the Mann–Whitney U-test; ** *p* < 0.01, **** *p* < 0.0001. (**C**) Representative histological cross sections of the colon at different time points during the experimental protocol. Left panel, wildtype mice; right panel, *Alox15b*-KI mice. Solid arrows indicate infiltrations of inflammatory cells. Dotted arrows indicate mucosal ulcerations.

**Figure 3 ijms-24-11034-f003:**
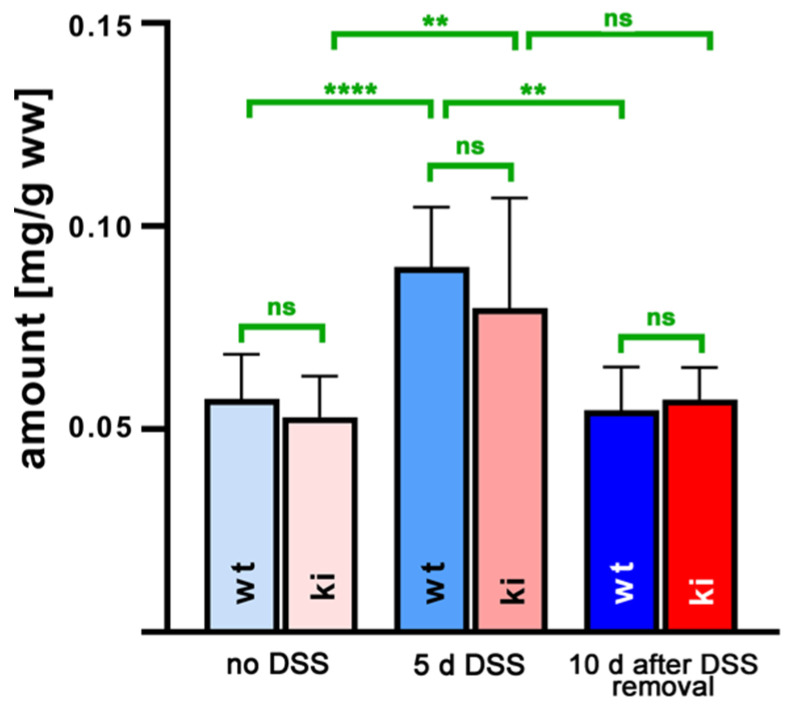
Quantification of hydroxy polyenoic fatty acids in colon tissue at different time points during DSS-induced colitis. Experimental colitis was induced in *Alox15b*-KI mice and outbred wildtype controls, as described in Section 4. At different time points, animals were sacrificed, colon was prepared, total lipids were extracted, extracts were hydrolyzed and the resulting free fatty acid derivatives were analyzed via LC-MS (see Section 4). The hydroxy fatty acids specified in Appendix A were separately quantified and summed up. Experimental raw data were evaluated via the Mann–Whitney U-test. Five representatives (*n* = 5) of the different experimental groups were analyzed. For the *Alox15b*-KI animals, 10 days after DSS removal, only 3 animals were analyzed (*n* = 3.) ns—statistically not significant, **—*p* < 0.01, ****—*p* < 0.0001.

**Figure 4 ijms-24-11034-f004:**
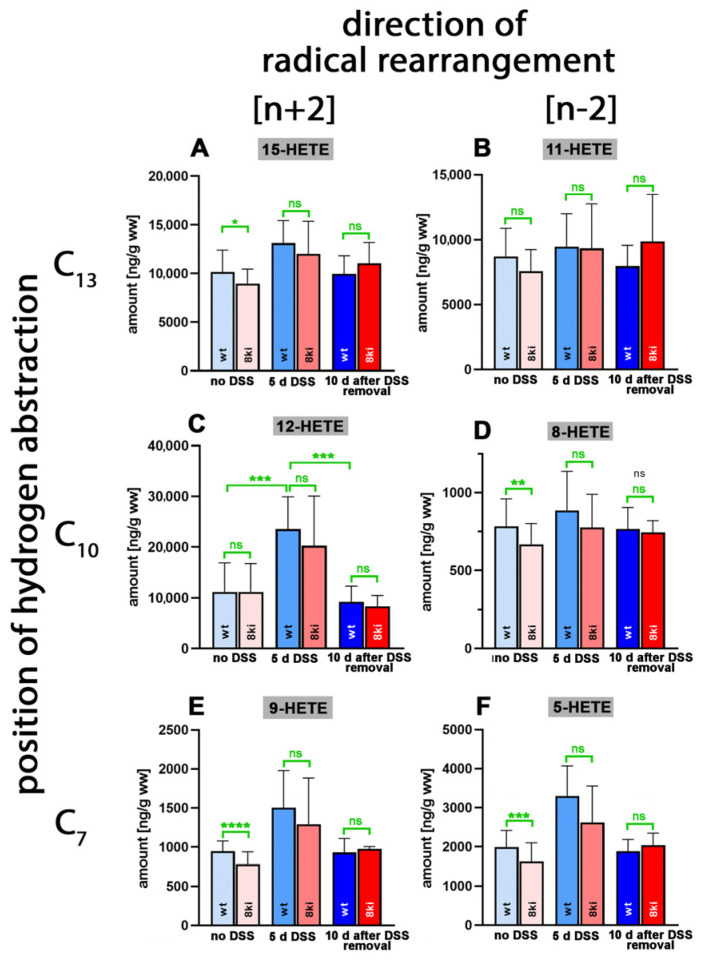
Quantification of arachidonic acid oxygenation products in colon tissue at different time points during DSS-induced colitis. Experimental colitis was induced in *Alox15b*-KI mice and outbred wildtype controls, as described in Section 4. At different time points, animals were sacrificed, colon was prepared, total lipids were extracted, extracts were hydrolyzed and the resulting free arachidonic acid oxygenation products were analyzed via LC-MS (see Section 4). The carbon atom, from which a hydrogen radical is abstracted during biosynthesis (**left** part of the image), and the direction of radical rearrangement ([*n* + 2] radical rearrangement in the direction of the methyl end of the fatty acid, [*n* − 2] radical rearrangement in the direction of the carboxylic group of the fatty acid) are indicated (**upper** part of the image). Experimental raw data were evaluated via the Mann–Whitney U-test. Five representatives (*n* = 5) of the different experimental groups were analyzed. For the *Alox15b*-KI animals, 10 days after DSS removal, only 3 animals were analyzed (*n* = 3). ns—statistically not significant, *—*p* < 0.05, **—*p* < 0.01, ***—*p* < 0.001, ****—*p* < 0.0001. (**A**) 15-HETE, (**B**) 11-HETE, (**C**) 12-HETE, (**D**) 8-HETE, (**E**) 9-HETE, (**F**) 5-HETE.

**Figure 5 ijms-24-11034-f005:**
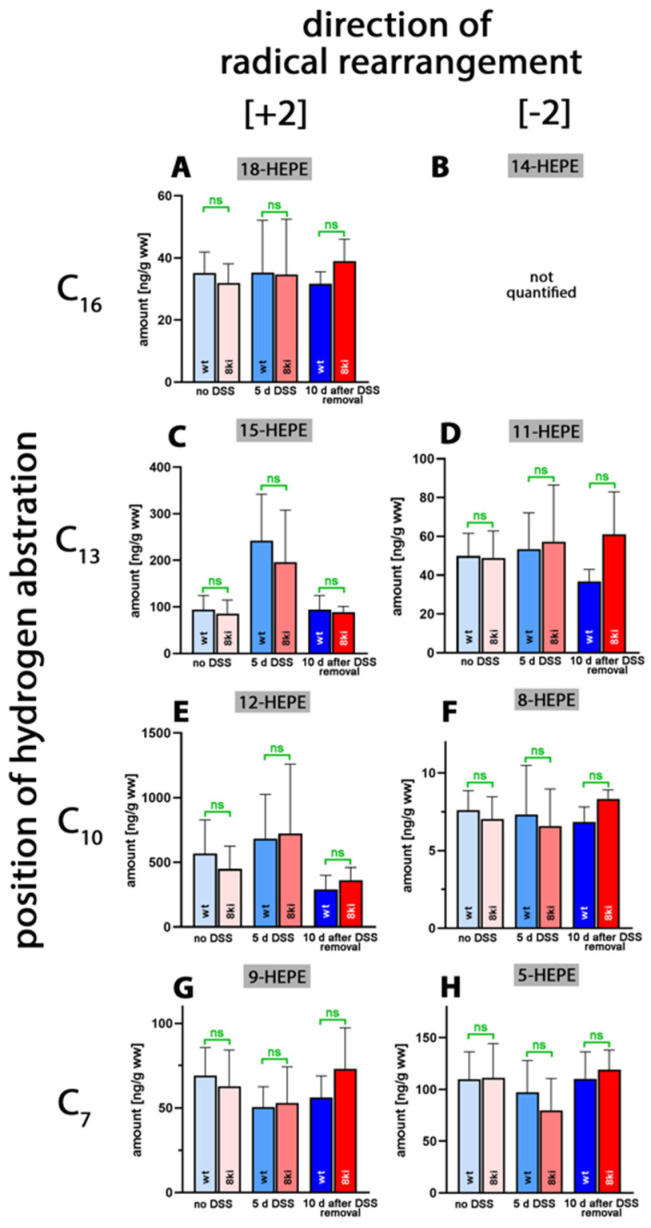
Quantification of eicosapentaenoic acid oxygenation products in colon tissue at different time points during DSS-induced colitis. Experimental colitis was induced in *Alox15b*-KI mice and outbred wildtype controls, as described in Section 4. At different time points, animals were sacrificed, colon was prepared, total lipids were extracted, extracts were hydrolyzed and the resulting free 5,8,11,14,17-eicosapentaenoic acid oxygenation products were analyzed via LC-MS (see Section 4). The carbon atom, from which a hydrogen radical is abstracted during biosynthesis (**left** part of the image), and the direction of radical rearrangement ([*n* + 2] radical rearrangement in the direction of the methyl end of the fatty acid, [*n* − 2] radical rearrangement in the direction of the carboxylic group of the fatty acid) are indicated (**upper** part of the image). Experimental raw data were evaluated via the Mann–Whitney U-test. Five representatives (*n* = 5) of the different experimental groups were analyzed. For the *Alox15b*-KI animals, 10 days after DSS removal, only 3 animals were analyzed (*n* = 3.) ns—statistically not significant. (**A**) 18-HEPE, (**B**) 14-HEPE (not quantified), (**C**) 15-HEPE, (**D**) 11-HEPE, (**E**) 12-HEPE, (**F**) 8-HEPE, (**G**) 9-HEPE, (**H**) 5-HEPE.

**Figure 6 ijms-24-11034-f006:**
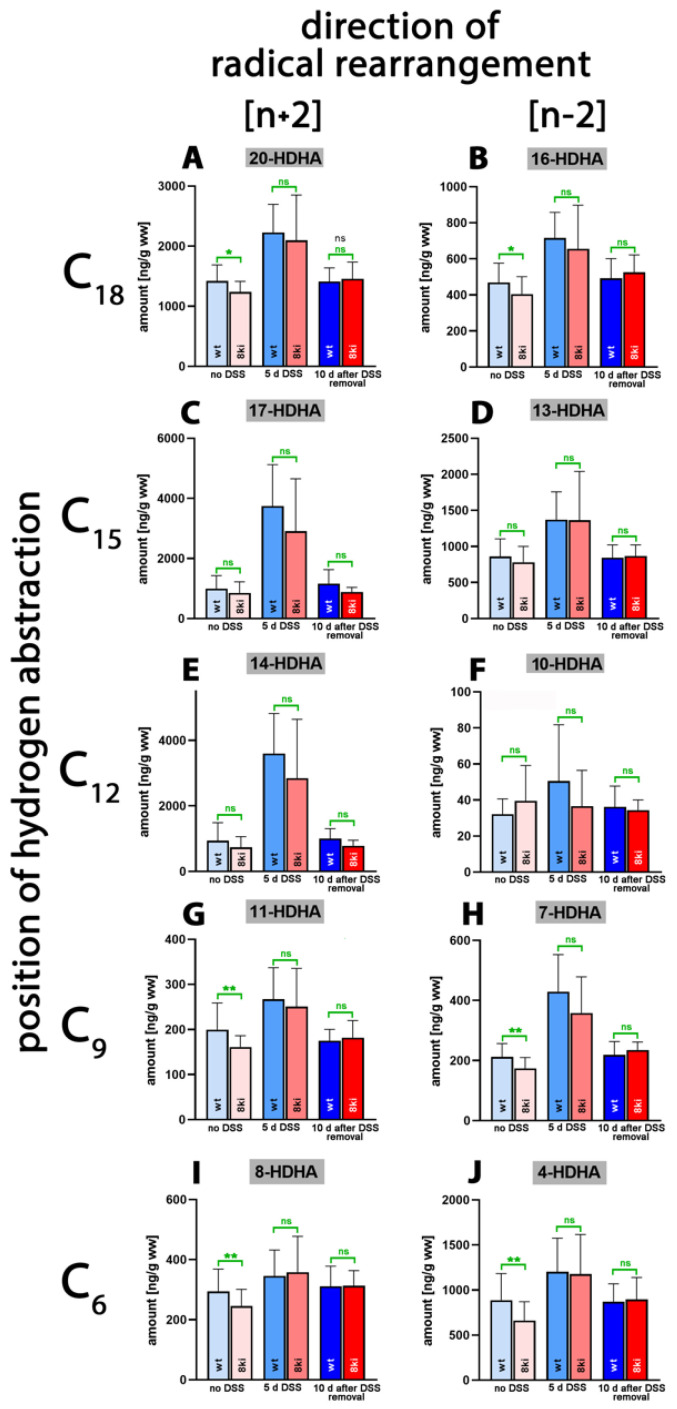
Quantification of docosahexaenoic acid oxygenation products in colon tissue at different time points during DSS-induced colitis. Experimental colitis was induced in *Alox15b-*KI mice and outbred wildtype controls, as described in Section 4. At different time points, animals were sacrificed, colon was prepared, total lipids were extracted, extracts were hydrolyzed and the resulting free docosahexaenoic acid oxygenation products were analyzed via LC-MS (see Section 4). The carbon atom, from which a hydrogen radical is abstracted during biosynthesis (**left** part of the image), and the direction of radical rearrangement ([*n* + 2] radical rearrangement in the direction of the methyl end of the fatty acid, [*n* − 2] radical rearrangement in the direction of the carboxylic group of the fatty acid) are indicated (**upper** part of the image). Experimental raw data were evaluated via the Mann–Whitney U-test. Five representatives (*n* = 5) of the different experimental groups were analyzed. For the A*lox15b*-KI animals, 10 days after DSS removal, only 3 animals were analyzed (*n* = 3). ns—statistically not significant, *—*p* < 0.05, **—*p* < 0.01. (**A**) 20-HDHA, (**B**) 16-HDHA, (**C**) 17-HDHA, (**D**) 13-HDHA, (**E**) 14-HDHA, (**F**) 10-HDHA, (**G**) 11-HDHA, (**H**) 7-HDHA, (**I**) 8-HDHA, (**J**) 4-HDHA.

**Figure 7 ijms-24-11034-f007:**
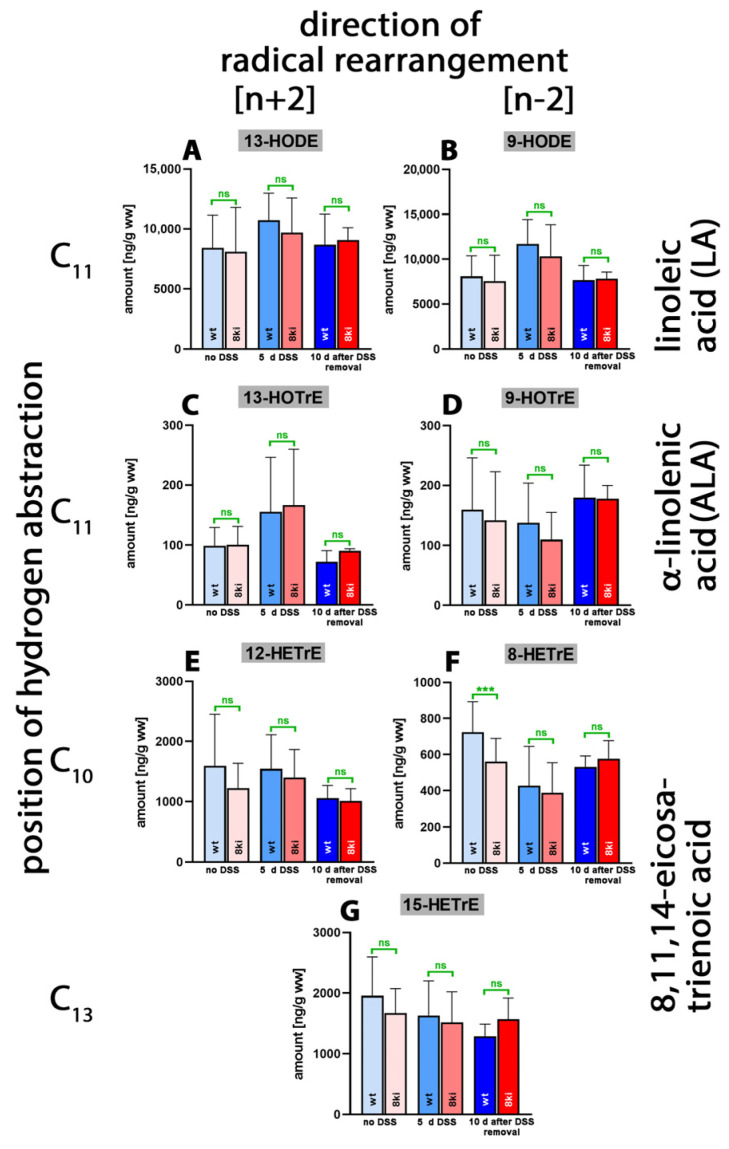
Quantification of the oxygenation products of different polyenoic fatty acids in colon tissue at different time points of DSS-induced colitis. Experimental colitis was induced in *Alox15b*-KI mice and outbred wildtype controls, as described in Section 4. At different time points, animals were sacrificed, the colon was prepared, total lipids were extracted, extracts were hydrolyzed and the resulting free polyenoic fatty acid oxygenation products were analyzed through LC-MS (see Section 4). The carbon atom, from which a hydrogen radical is abstracted during biosynthesis (**left** part of the image), and the direction of radical rearrangement ([*n* + 2] radical rearrangement in the direction of the methyl end of the fatty acid, [*n* − 2] radical rearrangement in the direction of the carboxylic group of the fatty acid) are indicated (**upper** part of the image). Experimental raw data were evaluated via the Mann–Whitney U-test. Five representatives (*n* = 5) of the different experimental groups were analyzed. For the *Alox15b*-KI animals, 10 days after DSS removal, only 3 animals were analyzed (*n* = 3). ns—statistically not significant, ***—*p* < 0.001. (**A**) 13-HODE, (**B**) 9-HODE, (**C**) 13-HOTrE, (**D**) 9-HOTrE, (**E**) 12-HETrE, (**F**) 8-HETrE, (**G**) 15-HETrE.

**Figure 8 ijms-24-11034-f008:**
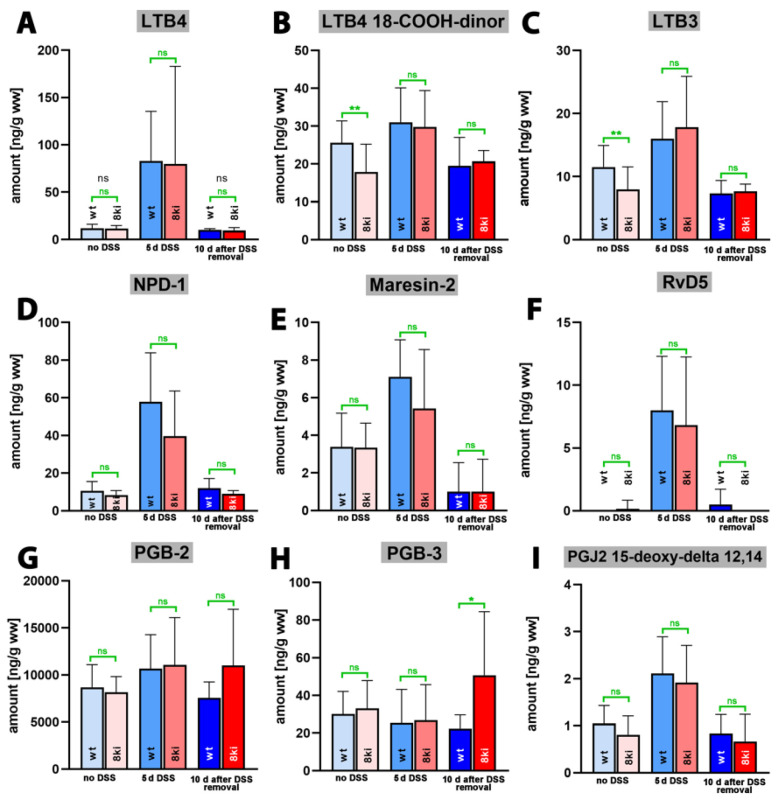
Quantification of complex oxygenation products of different polyenoic fatty acids in colon tissue at different time points of DSS-induced colitis. Experimental colitis was induced in *Alox15b*-KI mice and in outbred wildtype controls as described in Section 4. At different time points animals were sacrificed, colon was prepared, total lipids were extracted, extracts were hydrolyzed and the resulting complex polyenoic fatty acid oxygenation products were analyzed via LC-MS/MS (see Section 4). The carbon atom, from which a hydrogen radical is abstracted during biosynthesis (**left** part of the image) and the direction of radical rearrangement ([*n* + 2]—radical rearrangement in the direction of the methyl end of the fatty acid, [*n* − 2] radical rearrangement in the direction of the carboxylic group of the fatty acid) are indicated (**upper** part of the image). Experimental raw data were evaluated with the Mann-Whitney U-test. Five representatives (*n* = 5) of the different experimental groups were analyzed. For the *Alox15b*-KI animals 10 days after DSS removal only 3 animals were analyzed (*n* = 3). ns—statistically not significant. *—*p* < 0.05, **—*p* < 0.01. (**A**) LTB4, (**B**) LTB4 18-COOH-dinor, (**C**) LTB3, (**D**) NPD-1, (**E**) Maresin-2, (**F**) RvD5, (**G**) PGB2, (**H**) PGB3, (**I**) PGJ2 15-deoxy delta 12, 14.

**Figure 9 ijms-24-11034-f009:**
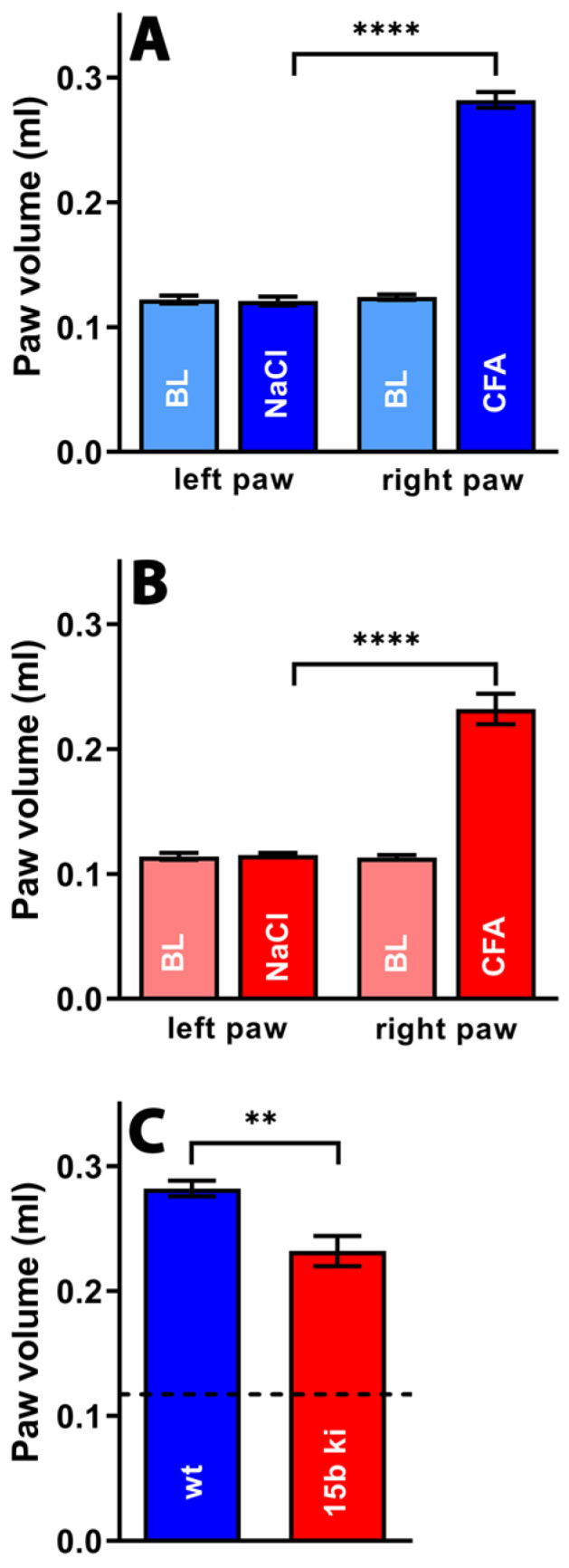
Freund’s complete adjuvant-induced paw edema inflammation model. Inflammation of mouse paw was induced via subcutaneous injection of Freund’s complete adjuvant, as described in the Section 4. After two days, the paw volume was measured (see Section 4) as the clinical readout parameter for the intensity of the inflammatory reaction. A Mann–Whitney U-test with *n* = 10 was performed in each experimental group. ns—statistically not significant; **** *p* < 0.0001, ** *p* < 0.01. (**A**) Comparison between paw volumes in wildtype mice. (**B**) Comparison between paw volumes in *Alox15b*-KI mice. (**C**) Comparison between paw volumes in wildtype and *Alox15b*-KI mice two days after CFA injection.

**Figure 10 ijms-24-11034-f010:**
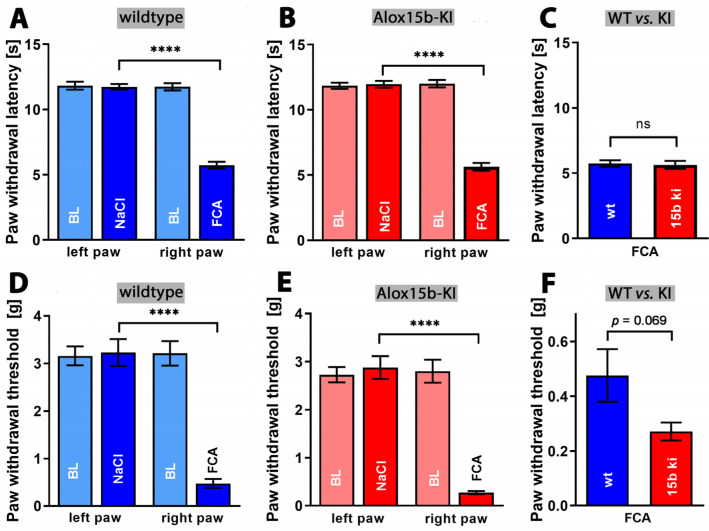
Quantification of pain perception in the Freund’s complete adjuvant-induced paw edema inflammation model. Inflammation of mouse paw was induced through subcutaneous injection of Freund’s complete adjuvant, as described in the Section 4. After two days, the paw withdrawal latency (see Section 4), Hargreaves test, panels (**A**–**C**) and the withdrawal threshold (see Section 4), von Frey test, panels (**D**–**F**) were measured as clinical readout parameters for pain perception. A Mann–Whitney U-test with *n* = 10 was performed in each experimental group. ns—statistically not significant; **** *p* < 0.0001. (**A**) Paw withdrawal latency of wildtype mice, (**B**) paw withdrawal latency of *Alox15b*-KI mice, (**C**) comparison between paw withdrawal latency of wildtype and *Alox15b*-KI mice two days after FCA injection (**D**) paw withdrawal threshold of wildtype mice, (**E**) paw withdrawal threshold of *Alox15b*-KI mice, and (**F**) comparison between paw withdrawal threshold of wildtype and *Alox15b*-KI mice two days after FCA injection.

**Table 1 ijms-24-11034-t001:** Quantification of additional dihydroxy derivatives of arachidonic and docosahexaenoic acids. Animal experiments, sample workups and LC-MS/MS analyses are described in the legend of Figure 8.

Genotype	Group	Metabolite (ng Metabolite/g Tissue Protein)		
		5*S*,12*S*-diHETE	8*S*,15*S*-diHETE	10*S*,17*S*-diHDHA
Wildtype (WT)	Day 0	38.3 ± 8.3	8.6 ± 1.2	24.3 ± 4.9
	Day 5	99.7 ± 32.7	16.4 ± 4.2	87.2 ± 41.3
	Day 15	31.3 ± 8.5	9.1 ± 3.0	16.4 ± 7.7
Knock-in (KI)	Day 0	35.2 ± 10.4	6.8 ± 1.0	14.6 ± 2.5
	Day 5	109.3 ± 79.3	18.9 ± 6.8	62.1 ± 33.4
	Day 15	32.2 ± 11.3	11.0 ± 0.7	12.4 ± 2.5

## Data Availability

All data generated or analyzed during this study are included in the published article and the online supplement Original experimental raw data can be obtained upon request to H.K. and D.H.

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
