# Peer review of "Humanization of the Reaction Specificity of Mouse Alox15b Inversely Modified the Susceptibility of Corresponding Knock-In Mice in Two Different Animal Inflammation Models"

_ijms, 2023, doi:10.3390/ijms241311034_

Round 1

Author Response

Please see appended PDF file!

Reviewer 2 Report

The article entitled "Humanization of the reaction specificity of mouse Alox15b in- 2 versely modified the susceptibility of corresponding knock-in 3 mice in two different animal inflammation models" is a well written article. The authors simply did a great work.

Author Response

Please see appended PDF file!

Reviewer 3 Report

The manuscript by Marjann Schäfer investigated the functional consequences of humanization of the reaction specificity of mouse arachidonic acid lipoxygenases 15b (Alox15b) in two different inflammation models using Alox15b knock-in mice. Alox15b knock-in mice (Alox15b-KI) are designed to express an AA 15-lipoxygenating enzyme variant (which is similar to human ALOX15B) instead of the AA 8-lipoxygenating wildtype enzyme (mouse Alox15b). The authors found that Alox15b-KI mice lost significantly more body weight during the acute phase of dextran sodium sulfate (DSS)-induced colitis, and recovered less rapidly during the resolution phase. On the other hand, Alox15b-KI mice were less susceptible to adjuvant-induced paw oedema when comparing to the wildtype mice. Apart from these changes, no other significant changes, e.g., colitis severity, colonic levels of oxylipin in colitis, pain perception in paw oedema, etc. were detected between Alox15b-K and wildtype mice. The difference responses to two inflammation models suggests the different pathophysiological roles of ALOX-isoforms in inflammation and resolution by producing pro-inflammatory or pro-resolving mediators in humans. Thus, this study will provide further understanding of the roles of ALOX-isoforms in inflammation pathways. Overall, the manuscript is well written, and the results are convincing. I would like to recommend the manuscript for publication after minor revision.

Minor comment:

In line 121, the author said “15-HETE was not detected”, however in figure 1, the signal was labelled as 15S-HETE. What is the difference between 15-HETE and 15S-HETE?

In figure2, the Alox15b KI mice lost more body weight, however, did not seem to exhibit more severe type of DSS-induced colitis. What could be the reasons to lose more body weight in Alox15b KI mice? Did the authors also study the weights of different organs and tissues?

In line 501, the authors suggest that Alox15b-KI mice experienced the more disruption of intestinal barrier during DSS-induced colitis. It could be better if the authors also study the tight junction proteins (e.g., ZO-1 and Occludin) in colon sections.

In figure 4, why 15-HETE level in colon tissue was significantly decreased in Alox15b knock-in mice, which is designed to express AA 15-lipoxygenating enzyme variant, and thus increase 15-HETE level?

Author Response

Please see appended PDF file!

Reviewer 4 Report

The experimental data exhibits high transparency and appears to have been conducted meticulously.

However, the paper merely reveals the absence of a certain factor in relation to the observed phenomenon and does not offer novel insights or groundbreaking discoveries.

Considering these factors, I have determined that it would be more appropriate to submit this manuscript to a journal other than IJMS.

This manuscript contains some typos. I recommend that you check it carefully.

Author Response

Please see appended PDF file!

Round 2

Author Response

Dear editor,

on behalf of all co-authors, I should like to thank the reviewers again for critical reading of our revised ms and for the helpful comments provided by reviewer 1. In response to her (his) criticism we modified the text and specified the configuration of the chiral centers of the quantified dihydroxy fatty acids analyzed for Table 1. We also modified the reference list according to the recommendation of the reviewer. In addition to the unlabeled version of the rerevised ms that can directly be used for the publication process we are submitting a labeled version, in which the new alterations introduced during the second round of revision are clearly labeled by green background. Alterations introduced during first revision are not labeled anymore.

Reviewer 1

Comment 1: Reference 48 on line 563 should be replaced or added to the previous one (Soares et al. BBRC 2005), which was the first report.

Response of authors: We follow the advice of the reviewer and added this reference (Soares et al, 2005) to the reference list.

Comment 2: Table 1: it should be precised whether 5,12-diHETE is the double lipoxygenase product of ArA or LTB4.

Response of authors: In the rerevised Table 1 we clarified that the quantified dihydroxy derivatives constitute the double oxygenation products by giving the configuration of their chiral centers (5S,12S-diHETE, 8S,15S-diHETE, 10S,17S-diHDHA).

Comment 3: This comment is also valid for 10,17-diHDHA which could be PD1/NPD1 or PDX, the latter being the double lipoxygenase product of DHA.

Response of authors: In the rerevised Table 1 we clarified that the quantified 10,17-diHDHA is the double oxygenation product of DHA (10S,17S-diHDHA).

Comment 4: Line 781: (NPD-1, NPDx) is confusing because PDX, the double lipoxygenase product of DHA is different from PD1/NPD1 that is a monooxygenase product of DHA, with its OOH intermediate which is converted into an epoxide further hydrolyzed by an epoxide hydrolase.

Response of authors: We apologize for this confusion. In response to the comment of the reviewer we slightly modified the text, which now reads: “In our lipidomic studies we analyzed the kinetic profiles of some putative pro-resolving metabolites (Figure 8), such as neuroprotectins (NPD-1, NPDx), maresin-2 (Mrs-2) and resolvin-D5 [5, 46] during the time course of DSS-induced colitis.”

We hope that the rerevised version of the ms is now acceptable for publication in IJMS.

Sincerely,

H. Kuhn
